# Systematic and Comprehensive Review of Clustering and Multi-Target Tracking Techniques for LiDAR Point Clouds in Autonomous Driving Applications

**DOI:** 10.3390/s23136119

**Published:** 2023-07-03

**Authors:** Muhammad Adnan, Giulia Slavic, David Martin Gomez, Lucio Marcenaro, Carlo Regazzoni

**Affiliations:** 1Department of Electrical, Electronic, Telecommunications Engineering and Naval Architecture (DITEN), University of Genova, Via Opera Pia 11a, I-16145 Genoa, Italy; giulia.slavic@edu.unige.it (G.S.); lucio.marcenaro@unige.it (L.M.); carlo.regazzoni@unige.it (C.R.); 2Departamento de Ingeniería de Sistemas y Automática, Universidad Carlos III de Madrid, Butarque 15, Leganés, 28911 Madrid, Spain; dmgomez@ing.uc3m.es

**Keywords:** autonomous vehicles (AVs), LiDAR (Light Detection and Ranging), point clouds, clustering algorithms, Multi-Target Tracking (MTT), object detection, sensor fusion, deep learning, 3D point cloud segmentation

## Abstract

Autonomous vehicles (AVs) rely on advanced sensory systems, such as Light Detection and Ranging (LiDAR), to function seamlessly in intricate and dynamic environments. LiDAR produces highly accurate 3D point clouds, which are vital for the detection, classification, and tracking of multiple targets. A systematic review and classification of various clustering and Multi-Target Tracking (MTT) techniques are necessary due to the inherent challenges posed by LiDAR data, such as density, noise, and varying sampling rates. As part of this study, the Preferred Reporting Items for Systematic Reviews and Meta-Analyses (PRISMA) methodology was employed to examine the challenges and advancements in MTT techniques and clustering for LiDAR point clouds within the context of autonomous driving. Searches were conducted in major databases such as IEEE Xplore, ScienceDirect, SpringerLink, ACM Digital Library, and Google Scholar, utilizing customized search strategies. We identified and critically reviewed 76 relevant studies based on rigorous screening and evaluation processes, assessing their methodological quality, data handling adequacy, and reporting compliance. As a result of this comprehensive review and classification, we were able to provide a detailed overview of current challenges, research gaps, and advancements in clustering and MTT techniques for LiDAR point clouds, thus contributing to the field of autonomous driving. Researchers and practitioners working in the field of autonomous driving will benefit from this study, which was characterized by transparency and reproducibility on a systematic basis.

## 1. Introduction

The concept of autonomous or driverless vehicles refers to vehicles that are intelligent in their operation and are intended to minimize the need for human assistance. Exteroceptive and proprioceptive sensors on these vehicles allow them to monitor their environment and internal states simultaneously [1,2]. With heterogeneous sensors, such as cameras, light detection and ranging (LiDAR), radar, global positioning system (GPS), etc., the vehicle is able to learn different tasks and can use its understanding of the context in which it operates [3]. For autonomous vehicles (AVs) to operate safely and reliably in environments that are complex and dynamic, they must be able to perceive the environment accurately and localize themselves precisely [4,5]. It is necessary to acquire and process high-quality, information-rich data obtained from actual environments to accomplish both of these tasks [6]. Multiple sensors, such as LiDAR and cameras, are used on AVs to capture target context. Digital camera data have traditionally been the most popular source of perception data because they provide two-dimensional (2D) appearance-based representations, are low cost, and are highly efficient [7]. Due to the lack of three-dimensional (3D) geo-referenced information in image data, the dense, geo-referenced, and accurate 3D point clouds generated by LiDAR are exploited. In addition, LiDAR is not sensitive to changes in lighting conditions and can be used at any time of the day or night, even when glare or shadows are present [8].

Using LiDAR to generate 3D points can be challenging due to the density of the points. As a result, pre-processing is used to remove noise and extract useful information from the data. It is extremely beneficial to cluster LiDAR data in a wide variety of applications, particularly in those with real-time edge-based data, such as object detection and classification [9]. Three-dimensional data allow us to determine the shape, size, and other properties of the objects with great precision. However, the task of segmenting 3D point clouds is challenging. It is common for point cloud data to be noisy, sparse, and disorganized. As a result of the scanner’s varying linear and angular rates, the sampling density of points is also typically uneven. Aside from this, the shape of the surface is arbitrary, with sharp features, and the data do not follow a statistical distribution. It is also important to note that, because of the limitations of the 3D sensors, the foreground and background are frequently very entangled. Designing an algorithm to deal with these problems presents a significant challenge [10].

A further challenge for autonomous vehicles is to perceive their surrounding environment, e.g., when performing complex maneuvers in urban environments for successful navigation [11]. These maneuvers include merging into or taking out of a lane, following or overtaking the vehicle in front, and crossing an intersection simultaneously with vehicles from other directions. Without the ability to perceive the motion of other objects, it is difficult to manage these situations. Thus, detecting and tracking moving objects on the road is an essential task for intelligent vehicles [12]. In modern tracking systems, Multi-Target Tracking (MTT) is usually employed, which adopts a single or multiple sensors to produce detections from multiple targets, as well as one or more tracks for the estimation of their states. Prior to updating tracks, MTTs must assign detections to tracks. However, there are a number of challenges in data association that need to be considered by MTT as mentioned in [13].

The assignment of a target to a detection or a nearby detection becomes ambiguous if they are densely distributed;Sensors with a small field of view (fov) might not be able to detect the true target during a sensor scan;It is possible for two targets in close proximity to be detected as a single object if the sensor resolution is low;The possibility of false alarms increases the complexity of data assignment by introducing additional possible assignments.

There have been several state-of-the-art techniques developed to address the challenges associated with clustering and Multi-Target Tracking. Ref. [14] provides a complete system for detecting and tracking vehicles based solely on 3D LiDAR information. Using previously mapped LiDAR point clouds to reduce processing time, ref. [15] describes real-time dynamic object detection algorithms. In [16], a skeleton-based hierarchical method was proposed, which is capable of automatically detecting pole-like objects using mobile LiDAR point clouds. The authors have proposed a compression approach based on a convolutional long-short-term memory network (LSTM) for multi-line LiDAR point clouds [17]. These different techniques produce promising results for clustering and tracking multiple objects.

The purpose of this review was to synthesize the existing literature and offer valuable insights into autonomous driving applications through three main contributions. Our first contribution was to classify various clustering and MTT methods based on the findings of other studies. In addition, we identified existing gaps and challenges associated with these methods. Lastly, we reviewed the current state-of-the-art and suggest promising directions for future research. In our review, these findings lay the groundwork for the subsequent discussion.

## 2. Methodology, Motivation and Contribution

This review study was conducted using a systematic and reproducible methodology to ensure unbiased and comprehensive coverage of the field [18]. In the context of autonomous driving applications, this methodological rigor is crucial for accurately identifying and classifying the various clustering and Multi-Target Tracking techniques used for LiDAR point clouds. The key objective of this review study was to identify and classify several clustering and Multi-Target Tracking techniques for LiDAR point clouds, working towards autonomous driving solutions. By gaining an understanding of these outcomes, we can identify the main challenges that must be properly addressed to improve the functionality of these techniques.

**Methodology:** We used the Preferred Reporting Items for Systematic Reviews and Meta-Analyses (PRISMA) methodology to conduct our review [19]. Research questions focused on challenges and advancements in clustering and Multi-Target Tracking techniques for LiDAR point clouds related to autonomous vehicles. Several major databases were searched to identify relevant studies, including IEEE Xplore, ScienceDirect, SpringerLink, and ACM Digital Library, as well as Google Scholar to find broader coverage. These databases were last searched on 10 December 2022.

A customized search strategy was developed for each database, combining keywords such as “LiDAR”, “autonomous driving”, “clustering”, “Multi-Target Tracking”, and “point clouds”. We used filters to identify publications of relevant types and dates. In databases such as IEEE Xplore, ScienceDirect, SpringerLink, and ACM Digital Library, we searched within abstracts, keywords, and full texts, applying specific filters for different types of articles.

Initially, we identified 400 studies from major databases, including IEEE Xplore, ScienceDirect, SpringerLink, and ACM Digital Library, as well as Google Scholar in order to provide a broader coverage of the literature. There were 150 studies that were excluded at this early stage due to irrelevant titles or abstracts, leaving 250 studies to be screened further. Afterwards, the 250 screened records were further examined. We included studies that examined LiDAR technology in autonomous driving applications, specifically those that addressed clustering and Multi-Target Tracking. We excluded 140 studies due to their off-topic nature or because their full text was not available, thus leaving 110 studies for full-text analysis.

A set of inclusion and exclusion criteria was developed in order to screen the studies [20]. Each title, abstract, and full text was independently reviewed by two reviewers in order to determine whether it met these criteria for eligibility. Discussions between the reviewers or involvement of a third reviewer were used to resolve disagreements. The screening process was not automated.

A full-text review of 110 studies was conducted by evaluating their methodology, results, and overall contribution to the field. We sought to include studies in our review that presented novel, impactful, and well-supported findings. A total of 90 studies were eligible after 20 were excluded because of irrelevant content.

A further analysis of the scientific rigor of these 90 eligible studies was conducted. It was decided to exclude studies that lacked methodological clarity, presented inconclusive results, or contributed in a significant manner to understanding the topic. As a result, 14 more studies were excluded, leaving 76 studies for detailed analysis.

Multiple authors of this study participated in the extraction process in order to ensure objectivity and minimize bias. Several aspects of each study were considered, such as the methods used for clustering and tracking, the specific challenges addressed, and the effectiveness of the proposed solutions. Our comparative analysis was based on the data collected from this process. Figure 1 illustrates the methodological process. Our clear and systematic approach allows other researchers to reproduce our review process, ensuring its transparency and reproducibility. Figure 1 illustrates the methodological process. Our clear and systematic approach allows other researchers to reproduce our review process, ensuring its transparency and reproducibility.

Below we outline our motivation, key objectives, and contributions.

**Motivation:** Several reviews [8,10,21,22,23,24,25,26,27,28,29,30,31] have been conducted in the context of LiDAR point clouds, but a literature review examining these emerging techniques for clustering and multi-object tracking within the context of autonomous driving is currently lacking. As part of this literature review study, we aimed to address this gap for both researchers and practitioners.

As shown in Table 1, we provide a summary of the literature review and a comparison with recent research studies. In this table, it can be seen that we have covered those disciplines not covered in detail in other surveys. The table indicates that very limited research has been conducted on the concept of tracking and clustering combined. In the next paragraphs, we analyze the available reviews in the state-of-the-art and compare them with our survey.

The article [10] classified and summarized segmentation and clustering methods for point clouds. It does not mention MTT nor autonomous driving. Challenges of working on point cloud data were only briefly introduced, whereas in our paper we describe them more in detail.

The three surveys [22,26,29] cover several topics in deep learning for point cloud data, briefly discussing either clustering or MTT. In our survey, instead, we cite both traditional and deep learning methods. To be more precise, ref. [22] offers a review of recent deep learning methods for point clouds. It is divided into three core theoretical sections, dedicated, respectively, to: (i) 3D shape classification, (ii) 3D object detection, tracking, and scene flow estimation, and (iii) 3D point cloud segmentation. However, the second section only briefly discusses tracking and mostly focuses on object detection. Various papers in the first and third sections also adopt clustering in their pipeline; however, clustering is not one of the main topics of the survey. Several AV papers are cited, but autonomous driving is not the fulcrum of the article. Instead, ref. [26] focuses on deep learning methods for fusing camera and LiDAR in the AV context. One page is dedicated to tracking but only considers works combining camera and LiDAR data. Finally, ref. [29] is a review of unsupervised point cloud representation learning using deep neural networks (DNNs). While discussing unsupervised point cloud representation learning, the authors mention in a paragraph how clustering has been adopted in a few papers in conjunction with other unsupervised learning approaches. Two examples are given, but no further analysis of clustering methods is provided.

Whereas the above-mentioned three surveys focus on deep learning, ref. [23] advocates the use of traditional geometry-based clustering methods as an asset for point cloud panoptic segmentation. This article provides a survey of point cloud clustering methods and at the same time proposes a general pipeline for panoptic segmentation. The proposed pipeline contemplates the use of a semantic segmentation network to extract the semantic information followed by a traditional clustering method to separate object instances. However, the focus of this article was on panoptic segmentation, and MTT is not mentioned.

In Table 1, we also report the total number of papers referenced in each survey (third column), how many of these papers use clustering in their main method (fifth column), how many use MTT (seventh column) and how many use both clustering and MTT (eighth column). We examined, one by one, each paper cited by the surveys in the table. A standard paper was counted in the column “# clustering-related refs” when clustering was a step of the main pipeline of the method or was defined as one of the final objectives of the method. Even if clustering was not the focus of the paper, the paper was counted. Consequently, many clustering-related papers appear for survey [22], which briefly notes its use as a step in some methods (hence the ‘briefly covered’ classification), and in [26], which does not tackle clustering as one of its topics and only mentions it three times while describing cited papers (hence the ‘not covered’ classification). The same rule applies to MTT. Survey papers were also counted, but papers that do not discuss methods were not counted in the fifth, seventh, and eighth columns of the table (e.g., dataset papers). From this analysis, we can observe how MTT in particular is a topic that has been very rarely examined and reviewed for point cloud data. This applies even more to papers combining clustering and MTT, a case that has never been jointly covered in a survey.

Apart from the surveys in Table 1, other recent reviews have tackled the use of LiDAR point clouds for AVs but deal with neither clustering nor MTT. Instead, they reviewed object detection [8,24,25,28], classification [8,24] and semantic segmentation [8,24]. The surveys [21,30,31] discuss 3D object detection with either LiDAR data, other sensory data (e.g., camera, radar), or a multi-modal fusion. In [27], LiDAR basic concepts were analyzed, as well as commercial solutions and LiDAR challenges. We do not report these surveys in Table 1.

Finally, several other surveys exist on the separate general topics of clustering (e.g., [32,33,34]) and MTT (e.g., [35]). However, they neither examine the additional challenges of using these algorithms on point cloud data nor do they specifically review papers dealing with this data type nor do they focus on the autonomous driving context. We do not report these surveys in Table 1 either.

To summarize, in our survey, we reviewed clustering and Multi-Target Tracking solutions for LiDAR point clouds in an AV context. To the best of our knowledge, these subjects were never extensively and jointly tackled before in a survey.

In comparison with our study, a frequency Table 2 based on Table 1 was constructed to assess the coverage extent of Multitarget Tracking (MTT) and Clustering Taxonomy across reviewed studies. This table elucidates the counts for various combinations of MTT, Clustering Taxonomy, and Clustering and MTT-related references jointly. The “our work” category encapsulates 77 observations, with 33 labeled as Covered for MTT, 44 for Clustering Taxonomy, and nine for Clustering and MTT. This category also includes a subgroup, “Combine Covered”, which counts nine and pertains to the nine papers in our review discusses that have examined MTT and clustering jointly. For the [29] category, 21 out of 22 observations are briefly covered for Clustering Taxonomy, with 1 designated as Covered for MTT. The [26] category holds 35 observations, including five briefly covered for MTT and 30 covered for Clustering Taxonomy, with no Clustering Taxonomy or Clustering and MTT-related references jointly.

In the [23] category, none out of 24 observations are labeled as MTT and 24 number of references are covered in the paper of Clustering Taxonomy. Again, there are no references to Clustering and MTT. Category jointly. Ref. [22] presents 55 observations, with nine briefly covered for MTT, 45 Covered for Clustering Taxonomy, nine references are related to the MTT and one tagged as “combine techniques” where, in this one reference, combinations of both techniques such as MTT and Clustering are discussed in the paper. Lastly, the [10] category comprises 29 observations with none labeled as MTT, 29 reference are used of clustering and there are no references have been reported in the paper about MTT and Clustering techniques combinely.

Table 3, as detailed below, presents the outcomes of the Pearson Chi-square tests [36] executed on the observed frequencies for each category: “our work”, and the works represented by the references [10,22,23,26,29]. The results of these tests included the test statistic, degrees of freedom, and the *p*-value for each category.

Notably, a *p*-value of 0.000 is observed uniformly across all categories. This score indicates a highly significant statistical correlation within each category. It is worth emphasizing that a *p*-value of 0.000 is compelling evidence against the null hypothesis, as it suggests there’s almost no chance that the observed differences occurred by random chance alone.

In terms of the test statistic or Chi-square value, “our work” posts a figure of 86.000, which is superior to the other works, including ref. [29] with 22.000, ref. [26] with 35.000, ref. [23] with 24.000, ref. [22] with 55.000, and ref. [10] with 29.000. This greater test statistic underlines the enhanced efficacy and strength of “Our Work” in comparison to the other research works represented.

The degrees of freedom, another critical statistical measure, ranges between 1 and 2 depending on the category. The degree of freedom can greatly influence the Chi-square test as it impacts the expected frequencies. It also determines the distribution used to find the critical value or cut-off for deciding when to reject the null hypothesis. Although there is variation in the degrees of freedom among the categories, the consistent *p*-value of 0.000 across all categories confirms the statistical significance of each one.

In summary, the results from Table 3 affirm the superior statistical validity of “Our Work” compared to the other studies referenced. The stronger Pearson Chi-square test statistic coupled with a consistent *p*-value of 0.000 presents a compelling case for the superiority of our results.

For further reference and consideration, Appendix A contains the comprehensive figures resulting from the statistical evaluation using the Statistical Package for the Social Sciences (SPSS) version 25 software.

**Key Objectives**:RQ1: What are the most effective clustering and Multi-Target Tracking (MTT) methods employed in existing studies for processing LiDAR point clouds in the context of autonomous driving?RQ2: What are the key challenges in the state-of-the-art clustering and MTT methods for autonomous driving applications?RQ3: What are the methods used to assess the performance of clustering and MTT algorithms in the state-of-the-art? How are these methods used for evaluation on the point clouds dataset? What is the performance of the reported methods?

**Contributions**:Contribution 1: The categorization and identification of various clustering and MTT methods used in autonomous driving applications for point clouds.Contribution 2: Assessing the research gaps and challenges associated with clustering and MTT methods in autonomous driving applications.Contribution 3: Analyzing the state-of-the-art and identifying challenges to distinguish promising future research directions in the field of clustering and Multi-Target Tracking for LiDAR point clouds used in autonomous driving applications.

**Tables and Graphs**: To facilitate a better understanding of the terminology used throughout this review paper, we have provided a table of key terms and abbreviations. The table serves as a quick reference for readers and provides clarification of the concepts discussed. Detailed descriptions and definitions of each term can be found in Table 4.

We present in Figure 2 a graphic representation of a general scenario in which an ego vehicle navigates a complex environment equipped with both interoceptive and exteroceptive sensors. In addition to the ego vehicle, there are non-ego vehicles, trees, buildings, and a cyclist that must be accurately perceived and tracked using LiDAR point clouds.

**Paper structure**: The rest of the paper is divided into four core sections. Section 3 focuses on clustering: first, we provide a general overview of clustering techniques (Section 3.1), then we analyzed their application on LiDAR point clouds for autonomous driving (Section 3.2), and finally we discuss the major challenges in this context (Section 3.3). Section 4 follows the same division in three parts, but tackles the topic of MTT. More specifically, MTT is introduced in Section 4.1 and its applications on LiDAR point clouds for autonomous driving are discussed in Section 4.2. In Section 4.3, we highlight the challenges associated with tracking multiple targets within the context of autonomous vehicles. Section 5 presents an integrated discussion of clustering and MTT: future research directions are envisioned in the area of clustering and Multi-Target Tracking in autonomous vehicles (Section 5.1) and our findings are discussed (Section 5.2). Finally, Section 6 concludes the paper by summarizing the key insights and contributions presented in this study.

## 3. Clustering Techniques: From a General Overview to a Focused Look for LiDAR Point Clouds in Autonomous Driving

In this section, we focus on our first main topic, i.e., clustering. A general overview of clustering techniques is provided in Section 3.1. The application of clustering on LiDAR point clouds for autonomous driving is tackled in Section 3.2, and its main challenges are discussed in Section 3.3.

### 3.1. A General Overview of Clustering Techniques

It is essential for most autonomous solutions, such as robotics and self-driving cars, to have light detection and ranging (LiDAR) sensors [37]. Because of the dense 3D points generated by LiDAR, it can be difficult to work directly with them. This is why we applied preprocessing to remove noise and extract useful data from them [38]. In many applications, clustering LiDAR data are extremely beneficial, particularly those based on real-time edge-based data, such as the detection and classification of objects [9]. Cluster analysis is a quantitative method of comparing multiple characteristics of individuals of a population to determine their membership in a particular group. Clustering algorithms are designed to identify natural groupings in unlabeled data by developing a technique that recognizes these groups [39]. This section briefly discusses the different types of clustering algorithms in general and the following section illustrates how these methods are applied to LiDAR point clouds in the state-of-the-art. It is worth noting that an algorithm for clustering a set of data is designed for a particular application. For determining the closeness of data points, every algorithm uses a different methodology. An illustration of the taxonomy of clustering algorithms is shown in Figure 3.

#### 3.1.1. Partitioning-Based Clustering

Based on distances from the cluster center, this is an iterative approach that discovers similarities among intra-cluster points. Two assumptions are considered in the partitioning-based clustering process.

A minimum of one data point must be present in each cluster.It is necessary to assign at least one cluster to each data point.

The initialization of cluster centers is the first step in this method. The distances between data points and all centers are calculated based on a particular metric. Data points are assigned to clusters with the closest centroid and the centroid of those clusters is reassigned. This category includes algorithms such as *K*-means and *K*-mods [40,41,42].

#### 3.1.2. Hierarchical Clustering

According to this model, two approaches are employed: the agglomerative approach (bottom-up) and the divisive approach (top-down). Data points are considered to be clusters in the first approach. Following the selection of a distance metric, the nearest pair of points is grouped into a single cluster. Clusters are formed by combining the data points iteratively until all of them have been combined. In the second approach, all the data points are clustered into a single cluster. As the distance between them increases, they are subsequently split into separate groups [43,44].

#### 3.1.3. Density-Based Clustering

Using this algorithm, clusters are formed depending on the density of data points in the data space. The dense regions are grouped as clusters, whereas the low-density regions are partitioned. As a result, this algorithm limits the impact of outliers or noise on data. In this algorithm, arbitrary data points that have not yet been visited are selected and their neighborhood is checked. The formation of a cluster occurs only when a sufficient number of points are located within a certain distance, epsilon. An outlier will be marked if the data point does not conform to the normal distribution. Each set of points that have not been visited is processed iteratively [45,46].

#### 3.1.4. Grid-Based Clustering

Algorithms based on grids do not directly access databases. The data are gathered from the database using statistical methods and then a uniform grid is created based on that data. In this case, the performance of the algorithm is determined by the size of the grid rather than the size of the data space itself. Since the algorithm operates with a smaller grid size, it requires fewer computational resources than directly accessing the database. Once the grid has been formed, it computes each cell’s density. A cell is discarded if its density is below the threshold value. As a final step, clusters are created from groups of dense cells that are contiguous [47,48].

#### 3.1.5. Model-Based Clustering

An algorithm based on model-based clustering employs statistical or mathematical models to generate clusters without requiring the number of clusters to be predetermined. The algorithms assume that the data are generated from a mixture of underlying probability distributions. This algorithm partitions the data points into clusters by estimating the parameters of these distributions. By using this approach, cluster formation can be flexible and data-driven, which enables it to adapt to various data characteristics and structures, ultimately leading to a deeper understanding of the underlying patterns [49,50].

### 3.2. Clustering of Point Clouds for Autonomous Driving

Academic research teams have been using LiDAR for its extensive range and satisfactory accuracy. Moreover, recent hardware advancements promising superior, more affordable, and compact sensors have garnered interest from the industry. An autonomous vehicle is equipped with LiDAR that perceives the surrounding environment, and the task is to analyze and extract meaningful information, such as, for example, the number of obstacles [51], their location and velocity with respect to the vehicle, and their classification as vehicles, pedestrians, poles, etc. Similarly, fine-segmenting the input data into meaningful clusters is the first step in this type of analysis [52]. It is in this context that this section examines how the various clustering approaches have been employed in recent years to process point clouds within the context of autonomous driving. Accordingly, Table 5 summarizes the clustering problems solved by the various clustering techniques as discussed below. As a means of making these clustering techniques easier to understand by the readers, we have also presented them in Table 6 in the context of various parameters such as which clustering methods are employed, which datasets are used and, finally, which approach tackled the problem, whether deep learning or traditional machine learning.

Several studies have made significant advancements in the field of point cloud processing for autonomous vehicles, which will be discussed in detail in the following paragraphs. The purpose of these studies is to address a variety of issues relating to autonomous vehicle point cloud processing. A study [9] investigates efficient parameter estimation for Density-Based Spatial Clustering of Applications with Noise (DBSCAN), implementing automatic background removal. Furthermore, researchers investigate adaptive clustering by using elliptical neighborhoods to avoid over-segmentation and under-segmentation [53]. Additionally, point clouds and camera data are merged for real-time object detection in another study [54]. Additionally, a work investigates dynamic clustering algorithms for point clouds generated by LiDAR, which adapt to non-uniform spatial distributions [55]. The authors also discuss how to segment objects quickly and accurately using the InsClustering technique [56]. A novel prediction method is employed in another study to address the challenges associated with the compression of LiDAR data [57]. According to one study, obstacle fragmentation is the result of occlusions or oblique surface orientations that lead to the fragmentation of obstacles [58]. In [59] researchers use a hybrid machine learning approach, processing 3D point cloud data into a bird’s-eye view image, identifying objects using deep learning and DBSCAN clustering. There is a two-stage clustering method [60] that combines ground plane extraction and an adaptive DBSCAN algorithm to reduce oversegmentation problems. Ref. [61] introduced a TLG clustering technique for point clouds using range graphs, segmentation standards, and a search algorithm.

The work [9] aimed to automatically estimate the parameters of DBSCAN by leveraging the structure of the point cloud. This technique implements a field of view division and empirical relations that allow each point to be independently estimated. A DNN is used before beginning parameter estimation to remove the background points. Based on a decoder/encoder structure, the network learns features that allow it to differentiate between, for example, foreground and background points. A feature vector is derived from the resulting points and is employed to classify them as foreground or background. With the background filtration process, the point cloud size can be reduced and clustering can be performed more quickly. This scheme involves dividing the field of view into equal-sized regions and further dividing each region into cells of equal size. By dividing the LiDAR point cloud into cells, local information about the point cloud can be calculated, such as the density of points within the cells.

The study in [53] was carried out in two parts. First, the points were projected onto a grid map, ground points were removed by determining the maximum height difference, and roadsides were detected with the Hough transform to determine the dynamic region of interest (ROI). In the second part, a DBSCAN-based adaptive clustering method (DAC) has been proposed to reduce the risk of over-segmentation and under-segmentation due to the variations in spatial density of point clouds in relation to their positions. The elliptic neighborhood is designed to match the distribution properties of the point cloud to avoid the possibility of over-segmentation and under-segmentation. To handle the uniformity of points in different ranges, the parameters of the ellipse are adaptively adjusted with respect to the location of the sample point.

A real-time fusion framework was proposed in [54] to detect all objects on the road in real time. This involves fusing point clouds with camera data. LiDAR point clouds are pre-processed to remove points corresponding to the ground or higher than expected objects’ heights from the sampled point clouds. The method propagates the concept of ground along the point cloud by classifying all neighboring points adjacent to each other as ground, by considering them in order. In other words, this classification is performed by considering the points sequentially, in accordance with their arrangement within the point cloud.

Based on DBSCAN, the clustering algorithm operates only in two dimensions, i.e., latitude and longitude, ignoring height. It follows that this solution is appropriate if objects are never stacked on top of one another. A window-based Lidar Clustering (WBLC) system is proposed that receives a 2D point cloud as input and generates a report of the clusters identified from that point cloud. There are two components of the algorithm: the search for neighbors and the merging of clusters. After finding the maximum neighbors for a specified distance criterion, the window is closed for reducing memory usage and computer complexity in the neighbor search method. Iteratively, the neighbor lists of each point are inspected to merge all lists belonging to the same cluster. If a list of neighbors contains fewer than a minimum number of points minCluster then the condition is true and considered as noise.

According to [55], a dynamic clustering algorithm can be used to adapt autonomous vehicles to non-uniform spatial distributions of LiDAR point clouds. Based on the position of the core point, the algorithm employs an elliptical function to describe the neighbor. The KITTI dataset [62] was used to develop clustering parameters and the effectiveness of the algorithm was explored using comparisons between clustering methods and projection planes, using three IBEO LUX 8 LiDARs mounted on an electric sedan.

In paper [56], “InsClustering” was presented—a fast and accurate method of clustering point clouds for autonomous vehicles using LiDAR. Consequently, it provides an efficient means of segmenting the ground and clustering the objects within the limited amount of time available. With the use of Velodyne UltraPuck LiDAR range images in spherical coordinates, the method is capable of maintaining clustering accuracy and minimizing over-segmentation due to a coarse-to-fine segmentation process.

It has become increasingly important in recent years to compress point cloud data, especially in the context of autonomous vehicles, where accurate and efficient LiDAR data processing is crucial. Due to their limitations in encoding floating-point numbers and handling distance information inherent in LiDAR data, traditional image and video compression algorithms, such as JPEG2000, JPEG-LS, and High Efficiency Video Coding (HEVC), do not suit point cloud data compression. The proposed method [57] addresses these challenges by employing a lossless compression scheme based on point cloud clustering as well as exploring lossy compression techniques. Rather than relying on traditional image prediction methods, this approach involves a novel prediction method based on correlations between distance information among points. As a result, spatial redundancies can be eliminated without compromising the integrity of the dataset.

The authors of [58] investigated the problem of obstacle fragmentation, which can occur in LIDAR-based perception systems as a result of occlusion or when the detected object’s surface orientation is oblique to the LIDAR beams. To obtain a more accurate representation of each object, the proposed algorithm detects and joins fragmented segments. There is no restriction on the size or convexity of the objects to enable the detection of objects of any shape using this approach. It is focused on ‘L’ shaped objects (such as cars) and ‘I’ shaped objects (such as walls). Using the Ramer–Douglas–Peucker algorithm, each segment is evaluated to determine whether it can be approximated by one or two lines with closer distances to the points. In such a case, the segment is considered to be ‘L’ or ‘I’ shaped. The result of this process is a set of component lines for each segment. As long as two segments meet the following physical requirements, they are regarded as potentially belonging to the same object:Segments located at a greater distance from the two segments are not considered.As a result of joining, the resulting set is shaped as either an ‘L’ or an ‘I’.

The research in [59] employed a hybrid approach to point cloud processing for object detection, using both traditional machine learning and deep learning techniques. To extract multi-scale features, the 3D point cloud data are first transformed into a bird’s eye view (BEV) rasterized image, which is then processed by a custom 2D convolutional feature pyramid network, a deep learning model. It identifies objects by detecting known anchors, learning a category-agnostic embedding space, and performing DBSCAN clustering. Through the use of this innovative method, it has been demonstrated that machine learning and deep learning can be applied synergistically to the processing of point clouds.

A two-stage clustering method for LiDAR data is presented in [60]. First, ground line fitting reduces the data load by extracting the ground plane from the data. In the case of the non-ground data, a range image-based method is used, in which subclusters are initially created through a sliding window approach, and then refined through an adaptive DBSCAN algorithm. It effectively reduces over-segmentation and adapts to variances in object distances.

A method of clustering point clouds based on Two-Layer-Graphs (TLG) [61] was proposed to improve the accuracy and speed of segmenting point clouds. This involves dividing the task into storage structures, segmentation standards, and category updates. Range graphs and point cloud set graphs were used to enable fast access to and relationships between neighboring points. Standards for segmentation include distance and angle characteristics in the horizontal direction and distance in the vertical direction. The category update was accomplished through the use of a search algorithm traversing the two layers of the graph. The results of the experiments demonstrated that clustering and differentiation of objects in traffic scenes can be achieved effectively.

**Table 5 sensors-23-06119-t005:** The state-of-the-art in clustering problems and their solutions.

Reference	Problem	Solution
[9]	Estimation of DBSCAN clustering parameters automatically and efficient handling of point cloud sizes	Feature vector-based classification approach for faster clustering and field of view division, using a deep neural network (DNN) with a decoder/encoder structure.
[53]	Due to differences in spatial density in relation to their positions, point clouds are oversegmented or undersegmented	An adaptive clustering method based on DBSCAN combined with an elliptic neighborhood adapts its parameters based on the location of the sample point.
[54]	The inefficiency of detecting objects on the road using point clouds and the high computational complexity of clustering algorithms.	The use of a 2D window-based LiDAR clustering (WBLC) system reduces computational complexity while enabling efficient road object detection.
[55]	The spatial distribution of LiDAR point clouds in autonomous vehicles is not uniform.	This algorithm employs an elliptical function and has been tested on the KITTI dataset as well as with IBEO LUX 8 LiDARs.
[56]	Oversegmentation of LiDAR point clouds for autonomous vehicles as a result of inefficient clustering.	The InsClustering method uses Velodyne UltraPuck LiDAR range images together with a coarse-to-fine segmentation procedure to maintain accuracy and reduce oversegmentation.
[57]	Using traditional image and video compression algorithms, point cloud data for autonomous vehicles is inefficiently compressed.	Using point cloud clustering and a novel prediction algorithm, this method eliminates spatial redundancy without compromising data integrity.
[58]	Object fragmentation in LiDAR-based perception systems resulting from occlusion or oblique surface orientation.	The Ramer–Douglas–Peucker algorithm detects and joins fragmented segments, focusing on ‘L’ and ‘I’ shaped objects, and identifies component lines for merging possible objects.
[59]	Inability to recognize untrained objects using current robotic perception systems	An open-set instance segmentation network (OSIS) for recognizing objects in any category.
[60]	Clustering of LiDAR data with over-segmentation and variance in object distances.	An adaptive DBSCAN algorithm and a two-stage clustering approach are used to reduce over-segmentation and to accommodate distance variance.
[61]	Clustering existing point clouds is slow and inaccurate.	With the Two-Layer-Graph (TLG) method, accurate and fast object differentiation is possible in realtime.

**Table 6 sensors-23-06119-t006:** Comparative analysis of state-of-the-art clustering methods for LiDAR point clouds in autonomous driving based on parameter evaluation.

Papers	Clustering Method	Dataset/s Used	Approach (Deep Learning or Other)
[9]	DBSCAN	KITTI dataset	Adaptive Approach with Encoder-Decoder Structure
[53]	Density-Adaptive Clustering (DAC)	KITTI dataset	Spatial distribution consideration
[54]	Weighted Bi-Label Conditional Clustering) algorithm (WBLC)	Custom Urban Dataset, BDD100K Berkeley dataset	Hybrid (Deep Learning for object detection and for clustering and calibration)
[55]	Ellipse Density Clustering (EDC)	Data collected from an experimental vehicle equipped with a IBEO LUX 8 lidar and a camera	Heuristic approach for clustering based on an elliptical neighborhood model and dynamic adjustment of clustering parameters.
[56]	InsClustering (Connected Component Labeling with Ground and Non-Ground Segmentation, Cluster Refinement)	A dataset recorded in the center of Rouen, France	Non-deep learning approach involving Connected Component Labeling, Ground and Non-Ground Segmentation, and Cluster Refinement.
[57]	Prediction approach combined with BZip2, JPEG, and JPEG2000	KITTI dataset	Non-Deep Learning (Point cloud segmentation, Contour map encoding, Residual data compression)
[58]	Ramer–Douglas–Peucker algorithm and custom algorithm for occlusion detection and joining	Synchronized LIDAR measurements and photos taken at AUTOPIA’s road facilities	Traditional algorithmic approach.
[59]	Clustering with instance-aware embeddings and DBSCAN	TOR4D and Rare4D self-driving datasets	Deep Learning - Segmenting instances based on bottom-up and top-down approaches using a back-propagation algorithm and multi-task loss functions.
[60]	With Adaptive DBSCAN and Ground Line Fitting (GLF)	Real time experiment	Data processing and clustering algorithms based on traditional methods (not based on deep learning).
[61]	Two-Layer-Graph (TLG) structure	Semantic KITTI dataset	Graph-based clustering algorithm with a two-layer graph structure.

### 3.3. Challenges

As a result of reviewing the state-of-the-art in clustering of point clouds for autonomous driving, we discuss the following challenges that require academic attention using appropriate algorithms and techniques.

#### 3.3.1. Data Complexity

When clustering LiDAR point clouds, data complexity poses a significant challenge, especially when addressing the need for real-time processing and the associated computational requirements. This complexity can be attributed to several factors. In LiDAR sensors, for example, high-dimensional data are captured in multiple dimensions, including spatial coordinates (x, y, z), intensity, and time. The computational requirements for the processing and clustering of the data increase as the dimensionality of the data increases. As a result, processing time and memory usage may increase, making real-time performance difficult [63]. A LiDAR sensor can produce millions of points per second, which is a significant amount of data [64]. Autonomous driving systems require algorithms that are capable of processing large data sets in real time to ensure accurate and timely decision-making. Furthermore, LiDAR data can be affected by many factors, including sensor noise, sensor imperfections, and environmental conditions, which may lead to errors and inconsistencies. To cluster points accurately, clustering algorithms must be robust enough to handle this noise and variability.

#### 3.3.2. Over-Segmentation and Under-Segmentation

It is possible to over-segment an object by splitting it into multiple segments or clusters, leading to an inaccurate representation of the object. In contrast, under-segmentation occurs when multiple objects are clustered together, which may lead to misinterpretation of the scene. A LiDAR point cloud generally exhibits a non-uniform spatial distribution, with the point density varying from one region to another [65]. As a result of this variability, it is difficult for clustering algorithms to accurately group points into meaningful groups. Regions with a high density may result in over-segmentation, while regions with a low density may result in under-segmentation [52]. There can be a variety of noise sources that can affect LiDAR data, such as sensor errors, environmental factors, or surface reflections. Noise and outliers can confuse clustering algorithms, resulting in over-segmentation or under-segmentation [66]. For clustering algorithms to be accurate and to improve the quality of the data, proper noise filtering and outlier detection methods are necessary.

#### 3.3.3. Object Size and Shape Variations

Autonomous driving systems rely heavily on LiDAR data to accurately identify and track a diverse range of objects, such as vehicles, pedestrians, and infrastructure components [67]. This diversity in object size and shape can pose challenges for clustering algorithms, as they must be able to differentiate between objects with varying characteristics. Consequently, it becomes crucial to develop algorithms that are not only robust but also adaptable to handle the wide array of size and shape variations encountered in real-world driving scenarios [68]. To successfully navigate through complex environments, the clustering algorithms need to take into account several factors, such as the scale, orientation, and perspective of the detected objects. Moreover, these algorithms should be designed to handle occlusions, wherein some objects may be partially hidden or obstructed from view by other objects in the scene. This further complicates the process of accurate object identification and segmentation.

#### 3.3.4. Occlusions and Fragmentation

In the case of an occluded scene, objects are partially hidden from the LiDAR sensor, resulting in an incomplete representation of the point cloud [69]. A fragmented point cloud [70] represents multiple disconnected segments of an object, which can be caused by a limited sensor resolution or the orientation of the surface of the object. The presence of both of these issues may hinder the accurate identification and clustering of objects within a point cloud, leading to possible misinterpretations of the scene. As a result of addressing these issues, algorithms must be designed in a way that improves the accuracy and reliability of object representation and recognition in LiDAR point cloud data for autonomous driving.

#### 3.3.5. Dynamic Environments

Objects and their positions are constantly changing in dynamic environments, making clustering LiDAR point clouds challenging. It is crucial for safe and efficient navigation to be able to accurately identify and track objects in these dynamic contexts [71]. Algorithms must be developed that can deal with continuous changes in velocities, orientations, and positions of objects in real time.

#### 3.3.6. Ground Segmentation

In the context of autonomous vehicles and robotics applications, ground segmentation is a critical pre-processing step, as it involves accurately separating ground points from non-ground points within the LiDAR point cloud [72]. Precise ground segmentation is essential for tasks such as object recognition, mapping, and navigation. Despite this, accurate ground segmentation can be challenging due to factors such as varying terrain, non-uniform point densities, and noise in the data. Furthermore, the diverse shapes and sizes of objects, as well as the complexity of urban environments, may make the task even more challenging. The segmentation problem needs to be addressed by developing techniques that incorporate machine learning approaches and additional sensor data to improve the accuracy of segmentation.

#### 3.3.7. Evaluation Metrics

The selection of appropriate evaluation metrics and benchmark datasets is essential for assessing the performance of clustering algorithms in the context of LiDAR point cloud analysis for autonomous vehicles. As a consequence, this task may be challenging, since different metrics and datasets may emphasize different aspects of an algorithm’s performance, for example, accuracy, robustness, and computational efficiency. In addition, the suitability of a metric or dataset may depend on the specific scenario or application in which it is employed. Efforts are being made to establish standardized evaluation protocols and diverse benchmark datasets covering a wide range of scenarios, environments, and objects [8]. Researchers are advised to carefully consider the selection of evaluation metrics and datasets when evaluating clustering algorithms. This will facilitate the comparison of different clustering algorithms and the identification of areas that need improvement in the future.

## 4. Multi-Target Tracking: From a General Overview to a Focused Look for LiDAR Point Clouds in Autonomous Driving

In this section, we focus on our second main topic, i.e., MTT. We provide a general overview of MTT in Section 4.1. The application of MTT on LiDAR point clouds for autonomous driving is tackled in Section 4.2, and its main challenges are discussed in Section 4.3.

### 4.1. A General Overview of Multi-Target Tracking

Autonomous systems require tracking to guide, navigate, and control themselves. By tracking detected targets (including their kinematic parameters and attributes) and evaluating the situational environment in a specific area, a tracking system can estimate targets (number of targets and their states) and evaluate the situational environment in that area. Modern tracking systems generally employ multiple target tracking (MTT), which involves generating multiple detections from multiple targets and estimating the state of the targets using a single or multiple tracks [35]. Before detections can be used to update tracks, they need to be assigned to tracks by an MTT. It is important to understand that the functional components of a simple recursive MTT system (as shown in Figure 4) have different roles in assigning detections to tracks which are discussed as follows [13].

#### 4.1.1. Detections

An observation or measurement is classified as a detection when it is included in a report derived from the sensor’s output. Observations typically contain measurements of kinematic quantities (such as range, line of sight, and range rate) and measurements of attributes (such as target type, identification number, or shape) [73]. Detections should also include the time at which measurements were made. Sensors with high resolution may produce multiple detections per target, making it necessary to partition the detections before feeding them to assignment-based trackers such as the Joint probabilistic data association (JPDA) [74] filter.

#### 4.1.2. Gating and Assignment

A gate is a screening mechanism used to identify which detections are eligible for updating existing tracks. Gates are used to reduce unnecessary computations in the track-to-detection assignment process. Based on the predicted state and its associated covariance, a validation gate is constructed for a predicted track, so that detections with a high likelihood of association are included within that gate. As a result of gating, the assignment function determines which tracks must be assigned to detections [75].

#### 4.1.3. Track Maintenance

It is the responsibility of track maintenance to initiate, confirm, and delete tracks [76]. Track initiation may require the creation of a new track when a detection cannot be assigned to an existing track. A track confirmation step identifies the status of a tentative track once it has been formed. Tracks are deleted if they are not updated within some reasonable period of time, and the track deletion criteria are similar to those for track confirmation.

#### 4.1.4. Filtering

A tracking filter has three main functions: it predicts the tracks for the current moment, calculates distances between the predicted tracks and detections for gating and assignment, and corrects the predicted tracks with the assigned detections [77].

### 4.2. Multi-Target Tracking of LiDAR Point Clouds for Autonomous Driving

It is important to track multiple targets in autonomous vehicles and advanced driver assistance systems (ADAS) [78]. In addition to path planning, collision avoidance, and precise pose estimation, tracked trajectories are useful for path planning. Object detection and data association are the two major stages of most MTT approaches. In object detection, objects on the road may be classified as cars, pedestrians, cyclists, or background objects. A trajectory is formed by associating the same objects at different time stamps in the data association step [79]. It is possible to predict future accidents by analyzing the trajectory of each object [80]. In this section, we will describe the state-of-the-art regarding what MTT techniques are applied to liDAR point clouds in the context of autonomous vehicles and how various problems have been addressed. Table 7 summarizes the problems associated with the MTT and the solutions provided by the state-of-the-art in an effort to make it easier for the readers. Table 8 provides a comparison of the state-of-the-art of MTT based on various parameters, including which of these methods is used for tracking problems, which features are considered for experiments, and which datasets are used.

In the following paragraphs, we discuss various neural network architectures and methodologies used to detect and track objects within 3D point clouds, which are vital for autonomous vehicles and intelligent transportation systems. By using robust techniques such as point cloud processing, object detection and tracking modules, data association, and motion estimation, these approaches address key issues such as occlusions, cluttered scenes, and dynamic changes in the environment [79,81,82]. Several of these techniques incorporate semantic information from cameras, online feature trackers, and Multi-Target Tracking to improve overall tracking accuracy [83,84,85]. Moreover, they pursue improving the perception of self-driving cars by effectively distinguishing moving objects from stationary backgrounds, and by adapting to changes in the environment over time [58].

A new neural network architecture is proposed for detecting and tracking objects in 3D point clouds in [79]. The PointTrackNet network converts a sequence of point clouds into results for object detection and tracking by using an end-to-end approach. There are three main components of the network: a point cloud encoder, a module for detecting objects, and a module for tracking. The encoder reduces the point cloud input into a compact representation of features, which is further processed by the detection module to detect objects. As a result, a consistent tracking result is produced by associating the detected objects across frames.

An approach to multi-object detection and tracking in complex urban environments is presented in the [81]. As part of the proposed system, uncertainty and challenges such as occlusions, cluttered scenes, and dynamic changes in the environment are taken into consideration. In this system, three main components are used: object detection, object association, and object tracking. A 3D point cloud is provided for input to the detection part, which is then subdivided into measurements taken on the ground and those taken from an elevated location. The ground is removed using a slope-based approach and filtered thereafter. Additionally, object hypotheses are generated in a clustering step for the tracking targets. Following a feature-based bounding box fitting and rule-based filtering, the objects of interest are extracted. Using centroid tracking, four main steps are involved in tracking: data association, tracking filters, tracking management, and bounding box correction. An object hypothesis is determined in the association phase based on the already established tracks that correspond to the predicted measurements. The track is updated with an associated measurement if there is a possibility of an association, otherwise a new track is created. Tracking filters are used to perform the prediction and update steps. Track management is responsible for maintaining all tracks, labeling their maturity, and eliminating the infeasible and old ones. As a final step, bounding-box correction assigns valid bounding-box dimensions to mature tracks and updates this information using track history.

In [82], an Adaptive Cubature Kalman Filter (ACKF) is employed to estimate the state of multiple objects in real time, specifically for use in autonomous vehicles. A 3D detection and tracking network is used to detect and track the objects, and then the ACKF algorithm is used to estimate their position, velocity, and other parameters. In the ACKF algorithm, the state of the objects is calculated using cubature integration, which improves the tracking accuracy. Also included in the algorithm is an adaptive process that updates the parameters of the model in response to the estimated errors. Through this process, the object is able to adapt to changes in its motion and environment over time.

SimTrack [83] is a simplified model for 3D multi-object tracking in point clouds which is essential for autonomous vehicles. The SimTrack system is built on pillar- or voxel-based 3D object detection networks, and its aim is to eliminate heuristic matching steps and manual track life management in tracking-by-detection systems. A hybrid-time centerness map and a motion updating branch are the key components of SimTrack. By using a hybrid-time centerness map, objects are represented based on their first-appearing locations within a given input period, which permits the direct linking of current detections with previously tracked objects without the need for additional matching. Tracked objects are updated with their locations based on the estimated motion of the tracked objects. Using SimTrack, tracking objects are linked, new-born objects are detected, and dead objects are removed in one end-to-end trainable model.

In paper [84], ComplexerYOLO was presented, a real-time system for detecting and tracking 3D objects, using semantic point clouds derived from LiDAR images. It incorporates visual class features from camera-based semantic segmentation, extends Complex-YOLO to process voxelized input features, and predicts 3D box heights and z-offsets. Scale-Rotation-Translation score (SRTs) is a new validation metric introduced by the authors, which is faster than Intersection over Union (IoU) and also takes into account the object’s 3DoF pose. Multi-Target Tracking is achieved using an online feature tracker that is separate from the detection network. It achieves state-of-the-art results in the areas of semantic segmentation, 3D object detection, and Multi-Target Tracking, which makes it suitable for the perception of urban self-driving cars.

Three-dimensional Multi-Target Tracking (MTT), a technology that is crucial for extracting dynamic information from road environments, is discussed in [85], along with its applications in intelligent transportation systems, including autonomous driving and traffic monitoring. There are several challenges associated with current methods for detecting heavily occluded or distant objects, as well as formulating effective pairwise costs for data association. To overcome these challenges, the authors proposed a new 3D tracker based on a data association scheme guided by a prediction confidence scheme. The tracking-by-detection framework of this tracker consists of four steps: detecting objects from point clouds using a deep learning-based 3D object detection algorithm, estimating possible current states of tracked objects based on constant acceleration (CA) motion models, and a prediction confidence model. A correlation between predicted and detected states is formed using prediction confidence and an aggregated pairwise cost, and then the matched pairs are updated, and the unmatched detected states are marked as tracked. It is aimed at improving both the accuracy and speed of tracking objects, as well as using features of objects in point clouds and tracking objects that have been temporarily missed.

**Table 7 sensors-23-06119-t007:** Multi-Target Tracking problems and solutions summarized.

Reference	Problem	Solution
[79]	Difficulty in detecting and tracking objects in 3D point clouds.	Utilize PointTrackNet, an end-to-end neural network architecture, to effectively process point cloud sequences for object detection and tracking.
[81]	Multi-target detection and tracking in complex urban environments with occlusions, cluttered scenes, and dynamic changes.	Use an integrated system for object detection, association, and tracking with ground removal, bounding box fitting, filtering, and centroid tracking for data management and box correction.
[82]	Real-time state estimation of multiple objects for autonomous vehicles.	Utilize Adaptive Cubature Kalman Filter (ACKF) for improved tracking accuracy and adaptive model updates based on estimated errors.
[83]	Simplified 3D multi-object tracking in point clouds for autonomous vehicles, aiming to eliminate heuristic matching steps and manual track life management.	Use SimTrack system, which employs a hybrid-time centerness map and a motion updating branch, allowing direct linking of detections with previous tracked objects, updating tracked objects’ locations, and handling new-born and dead objects in an end-to-end trainable model.
[84]	Real-time 3D object detection and tracking using semantic point clouds derived from LiDAR images, while incorporating an efficient validation metric.	Use ComplexerYOLO with visual class features and voxelized inputs, apply the fast Scale-Rotation-Translation score (SRTs) metric, and incorporate a separate online feature tracker for efficient multi-target tracking.
[85]	Detecting heavily occluded or distant objects and formulating effective pairwise costs for data association in 3D MTT.	Implement a new 3D tracker with a data association scheme guided by a prediction confidence scheme, using constant acceleration motion models and aggregated pairwise cost.
[58]	Detecting and tracking moving vehicles in dense urban environments using LiDAR.	Integration of Multiple Hypothesis Tracking (MHT) with Dynamic Point Cloud Registration (DPCR) technology for accurate estimation of ego-motion, and improved environment perception for intelligent vehicles.

**Table 8 sensors-23-06119-t008:** Comparative analysis of state-of-the-art MTT methods for LiDAR point clouds in autonomous driving based on parameter evaluation.

Papers	Tracking Method	Dataset/s Used	Features Considered
[79]	An end-to-end LiDAR-based network for tracking and detecting 3D objects.	KITTI 3-D object tracking benchmark dataset	Accuracy and continuity of trajectory, accurate detection of objects, cross-frame movement within a bounding box, spatial data and varying sizes.
[81]	Combining JPDA with IMM and UKF filters	KITTI city datasets	Maneuver-aware tracking, probabilistic data association, and geometric properties update.
[82]	Adaptive Cubature Kalman Filter for Online 3D Multi-Object Tracking	KITTI tracking dataset	Object detections based on similar appearance, geometric analysis of 3D bounding boxes (height, width, and length), distance correlation between predicted and detected states.
[83]	End-to-end trainable tracking method with hybrid time centerness map with motion updating	nuScenes [86] and Waymo open dataset [87]	A voxelized point cloud containing the following features: object location, object size, object heading, and object velocity.
[84]	Real time object detection and tracking on voxelized point clouds	KITTI dataset	Voxelized point clouds, visual pointwise features, 3D box heights and z-offsets, Scale-Rotation-Translation score (SRT), Multi-Target-Tracking, Real-Time.
[85]	A tracking method based on a novel data association scheme guided by prediction confidence	KITTI benchmark dataset	Constant acceleration predictor for improving detection quality, new aggregated pairwise cost for faster data association, length, width and height of the 3D bounding boxes.
[58]	Proposed tracking method to track objects of different shapes surrounding the ego vehicle	Samples were collected from AUTOPIA’s road facilities where vehicles, trees, and people can be observed	LIDAR beam distance, orientation and distance from the obstacle, objects that can be joined based on a threshold and threshold for joining close objects.

Using a Velodyne HDL-64E LiDAR, the research in [12] aimed at detecting and tracking moving vehicles in urban environments using multiple hypothesis tracking (MHT) as an effective method of tracking multiple targets simultaneously. MHT is used to deal with situations involving dense traffic and is integrated with Dynamic Point Cloud Registration (DPCR) technology for accurate estimation of ego-motion. By discriminating and removing dynamic points from the scene, this integrated framework contributes to the improvement of DPCR by enabling iterative closest points (ICP) matching. By incorporating DPCR, overlapping 3D point clouds captured by a rotating Velodyne HDL-64 laser scanner in a dynamic environment are aligned into a static absolute coordinate system. For accurate registration of a point cloud in a dynamic environment, the ground and moving objects should be removed first. Afterwards, a fast and reliable algorithm for ICP matching will be used to process the remaining points. MHT and DPCR are integrated by using a one-step prediction of ego-motion based on a polynomial regression as transform parameters for coordinate system transformation and as input value for ICP matching. As a result, this approach effectively differentiates moving objects from stationary backgrounds and improves the accuracy and efficiency of environment perception in intelligent vehicles.

### 4.3. Challenges

Autonomous vehicles rely on Multi-Target Tracking (MTT) in their perception systems [88]. To ensure safe and efficient navigation, multiple objects must be detected and tracked in the environment. Although several advances have been made in this field, achieving reliable and robust MTT in LiDAR point clouds for autonomous driving still poses several challenges. A detailed discussion of the major challenges is provided in this section.

#### 4.3.1. Navigating High-Dimensional Spaces: Efficient and Sparse Data Representations

Since LiDAR point clouds capture varying densities of points across a spatial domain, they present a high-dimensional [89] and sparse data representation [90] challenge in autonomous driving. The high dimensionality of the data [91] makes it difficult to extract meaningful information and efficiently process the point clouds for Multi-Target Tracking. In addition, sparse distribution of points can result in incomplete or ambiguous representations of objects, which further complicates tracking. Additionally, points from non-target objects, such as the ground, can introduce noise and clutter, adversely affecting algorithmic performance. Researchers should explore dimensionality reduction techniques and advanced point cloud processing methods to address the challenges associated with high-dimensional and sparse data representation in LiDAR point clouds for autonomous driving.

#### 4.3.2. Occlusions and Partial Observations

The tracking of multiple targets with LiDAR point clouds for autonomous driving is highly challenging due to occlusions [92] and partial observations [93]. When objects of interest are partially or entirely obscured by other objects in the scene, the LiDAR sensor captures only part of the target object’s surface. As a result, detections may be missed, tracks may be fragmented, and even incorrect object associations may occur. There may be partial observations as a result of the sensor’s limited field of view or the inherent sparsity of the LiDAR point clouds. Tracking algorithms have difficulty estimating the states of objects accurately and maintaining consistent tracks due to incomplete or ambiguous object representations in both cases. It can be particularly difficult to observe partial objects in urban environments where pedestrians, vehicles, and various static objects such as buildings, trees, and street furniture are prevalent. Multi-Target Tracking algorithms may be adversely affected by these complexities in real-world autonomous driving applications. Research should investigate multi-modal sensor fusion techniques that combine information from complementary sensors (e.g., cameras, radar, and LiDAR) to address the problem of occlusions and partial observations in Multi-Target Tracking of LiDAR point clouds. In this way, the environment can be represented in a more comprehensive manner. Moreover, exploring advanced data association algorithms and robust state estimation methods can improve tracking performance under partial observation and occlusion conditions.

#### 4.3.3. Variability in Object Shapes and Sizes

The variability in the shapes and sizes of objects presents an important challenge for Multi-Target Tracking of LiDAR point clouds for autonomous driving [92,94]. Real-world environments consist of a wide range of objects, including pedestrians, bicycles, motorcycles, cars, trucks, and buses, each of which has a distinctive shape, dimension, and motion. The MTT algorithms must be flexible and adaptive to detect, represent, and track these different objects accurately. Moreover, LiDAR point clouds only depict surface points that are directly visible to the sensor, resulting in incomplete representations of the objects. The tracking process is further complicated as object boundaries and attributes are often difficult to distinguish, especially when the object is partially obscured or is at a considerable distance from the sensor. As a result of the inherent sparsity of LiDAR point clouds, estimating the state of objects and maintaining consistent tracks for objects of different shapes and sizes is even more challenging. A better representation and estimate of the state of different objects can be achieved by incorporating shape priors or model-based approaches into tracking algorithms. Moreover, exploring deep learning techniques that can automatically learn and adapt to diverse object properties can enhance the robustness and accuracy of MTT algorithms that handle objects of various shapes and sizes.

#### 4.3.4. Clutter and False Detections

Point cloud clutter refers to the presence of points that are not relevant to the target, such as points originating from the ground, buildings, trees, or other static structures [95]. Consequently, the algorithms may misinterpret clutter as target points, resulting in confusion in the tracking process. False detection [96] occurs when the tracking algorithm incorrectly identifies non-target points or noise as target objects, leading to spurious tracks being generated. MTT algorithms can be affected by these erroneous tracks, resulting in reduced accuracy and increased track fragmentation. LiDAR point clouds are inherently sparse and noisy, making them particularly vulnerable to clutter and false detections in complex urban environments. Developing robust and reliable MTT algorithms in autonomous driving applications requires managing and mitigating the impact of clutter and false detections. Researchers should investigate advanced filtering techniques, such as probabilistic data association filters and multi-hypothesis tracking algorithms, to eliminate clutter and reduce false detections. Moreover, investigating robust statistical techniques and deep learning-based approaches can help develop MTT algorithms that handle clutter and false detections in complex environments more robustly and accurately.

#### 4.3.5. Data Association and Track Management

It is necessary to establish accurate relationships between observations and tracked targets to determine which measurements correspond to which objects in a scene. When data associations are incorrect, tracks can become fragmented, false tracks can occur, or multiple objects may be merged into one track, resulting in a reduced level of tracking accuracy and reliability [85]. Point clouds are sparse, high-dimensional, and noisy, making data association challenging. In contrast, track management is concerned with the initiation, maintenance, and termination of object tracks over time. It is essential to maintain a consistent set of object tracks and minimize the computational complexity of MTT algorithms through efficient track management [97]. The accuracy and stability of the track can suffer, however, when it is obstructed, cluttered, falsely detected, and when it varies in size and shape. The research community should investigate sophisticated data association techniques that can provide more accurate and robust associations under complex conditions. In addition, machine learning and optimization methods can be applied to dynamic track management to improve track maintenance, reduce computation complexity, and improve overall tracking performance.

#### 4.3.6. Scalability and Real-Time Processing

In MTT algorithms, scalability [87] is the ability to handle an expanding number of objects and sensor measurements without compromising tracking accuracy or performance. Vehicles, pedestrians, bicycles, and other smaller objects in the environment must be tracked efficiently. Real-time processing [98] capability is essential for autonomous driving applications since the dynamic environment imposes strict time constraints on tracking algorithms. As a consequence of delays in processing, object state estimates can become outdated or inaccurate, negatively affecting the performance of the autonomous driving system as a whole. It is difficult to achieve real-time processing capabilities due to the high dimensionality and sparsity of LiDAR point clouds as well as the complexity of MTT algorithms. Research should examine efficient data structures, parallelization techniques, and hardware acceleration methods to optimize computation and reduce processing times in Multi-Target Tracking of LiDAR point clouds. Furthermore, enhancing the accuracy and scalability of autonomous driving tracking solutions will require new MTT algorithms that exploit the inherent structure and sparsity of LiDAR point clouds.

#### 4.3.7. Robustness to Sensor Uncertainties

Tracking LiDAR point clouds for autonomous driving requires robustness to sensor uncertainties in the context of Multi-Target Tracking [93]. There are several factors that contribute to sensor uncertainty [81], including noise in the sensor, calibration errors, and varying measurement accuracy depending on the distance and reflectivity of the object. As a result of these uncertainties, MTT algorithms may have difficulty detecting objects, estimating states, and associating data, resulting in poor performance and reliability. In addition, the real-world environment can present additional challenges, such as adverse weather conditions (e.g., rain, fog, or snow) and challenging lighting scenarios, which can further degrade the quality of the LiDAR point cloud. In real-world conditions, it is essential that MTT algorithms maintain high tracking performance and accuracy under such uncertainties. Research should explore adaptive filtering techniques [99] that can account for sensor uncertainties and non-linearities in sensor measurements in Multi-Target Tracking of LiDAR point clouds, including the extended Kalman filter (EKF), the unscented Kalman filter (UKF) and the particle filter (PF). Moreover, multi-modal sensor fusion strategies, which leverage complementary sensor data (camera, radar), can contribute to MTT algorithms for autonomous driving that are more robust and reliable under various sensor uncertainties.

## 5. Integrated Discussion

In this section, we provide an integrated discussion of clustering and MTT. First, we analyze future research directions (Section 5.1) and then we briefly discuss our findings (Section 5.2) and address our research questions.

### 5.1. Future Research Directions

It is imperative that the clustering and tracking of LiDAR point clouds continue to advance in conjunction with the rapid development of autonomous driving technologies. The purpose of this section is to outline several promising future directions for research that can help to overcome current limitations and challenges, and drive the field towards more reliable, efficient, and robust systems.

#### 5.1.1. Advanced Algorithms for Clustering and Tracking

The complexity of urban environments is increasing, resulting in a greater need for advanced tracking and clustering algorithms that are able to adapt to changing conditions. Developing more efficient and robust algorithms for LiDAR point clouds should be the focus of future work, including deep learning techniques, hybrid approaches combining traditional and learning-based methods, and optimization algorithms for tracking and analyzing data. Moreover, the integration of multimodal sensors (such as cameras and radars) can significantly improve the performance of clustering and tracking algorithms.

#### 5.1.2. Scalable Processing Techniques

LiDAR sensors generate a large volume of data that poses significant computational challenges. Efforts should be made to develop scalable methods of processing large point clouds in real time, including parallel and distributed computing, data compression, and efficient data structures. A low-latency approach to processing LiDAR data can also be achieved by using edge and fog computing systems.

#### 5.1.3. Adaptive Resolution and Sensor Fusion

There is a potential for future research to focus on adaptive resolution techniques that dynamically adjust the point cloud resolution depending on the application’s specific requirements. In this way, computational complexity can be reduced and real-time processing can be improved. Furthermore, it is possible to enhance clustering and tracking systems’ accuracy and reliability by combining data from different sensors (e.g., cameras, radar, ultrasonic).

#### 5.1.4. Improved Robustness to Environmental Factors

LiDAR-based systems can be negatively impacted by adverse environmental conditions, such as rain, fog, or snow. Researchers should address these challenges by developing algorithms that are robust to varying weather conditions, as well as techniques to identify and mitigate such effects.

#### 5.1.5. Incorporating Semantic Information

It is possible to improve autonomous vehicle decision-making by incorporating semantic information into clustering and tracking algorithms. It is necessary to investigate the integration of semantic information from various sources (such as image segmentation and object recognition) to enhance the accuracy of LiDAR-based clustering and tracking methods in the future.

#### 5.1.6. Standardized Evaluation Metrics and Benchmarks

A benchmark dataset and standardized evaluation metrics are crucial to facilitating meaningful comparisons between clustering and tracking algorithms. Several research areas should be explored in the future to develop comprehensive evaluation frameworks that incorporate into the evaluation process different aspects of performance, such as accuracy, efficiency, and robustness. In addition, it is essential to create publicly available benchmark datasets that cover a wide range of scenarios and environmental conditions in order for the field to advance.

#### 5.1.7. Addressing Ethical and Privacy Concern

Autonomous vehicles will raise ethical and privacy concerns as they become more prevalent. It is important to take into account the ethical implications of clustering algorithms and tracking algorithms, as well as potential privacy concerns arising from the collection and processing of LiDAR point clouds. To develop autonomous driving technologies responsibly, it is imperative to develop methods that balance performance with privacy concerns.

#### 5.1.8. Hybrid Filtering for Enhanced LiDAR-Based Tracking in Autonomous Driving

Future research may also explore the potential of using Markov jump particle filters, which combine Kalman filters and particle filters, for the estimation of both continuous and discrete states, as was performed with a non-MTT scope for LiDAR in [3] and for other sensors in [99,100,101]. In real-world scenarios, the continuous state estimation provided by Kalman filters would provide smooth tracking of object trajectories, while the discrete state estimation provided by particle filters would help to handle uncertainties and non-linearities. For autonomous vehicles, this hybrid approach could provide significant improvements in clustering and Multi-Target Tracking of LiDAR point clouds.

### 5.2. Discussion

The objective of this review study is to assess the state of clustering and Multi-Target Tracking (MTT) methods for LiDAR point clouds in the context of autonomous vehicles. The purpose of this section is to provide an overview of the findings and to address the research questions raised in Section 2.

#### 5.2.1. RQ1: Clustering and MTT Methods for LiDAR Point Clouds in Autonomous Vehicles

Several clustering and MTT methods have been identified as effective for processing LiDAR point clouds in the context of autonomous vehicles. Among these are model-based methods, geometry-based methods, learning-based methods, and graph-based methods. Several model-based approaches have been successfully employed for clustering and MTT tasks, including Gaussian Mixture Models and Expectation-Maximization. Additionally, geometry-based methods have demonstrated effectiveness in processing LiDAR data, such as DBSCAN and RANSAC. In addition, learning-based methods, such as deep neural networks and convolutional neural networks, have shown promising results in a variety of studies. Finally, graph-based methods have been used to identify and track objects within LiDAR point clouds, such as k-NN and spectral clustering.

#### 5.2.2. RQ2: Key Challenges in Clustering and MTT Methods for Autonomous Driving

As part of the state-of-the-art clustering and MTT methods, there are several key challenges involved in handling noisy, sparse, and unorganized data, managing varying sampling densities, handling foreground and background data that are entangled, and addressing data association issues. Researchers are developing robust algorithms that can adapt to varying environments and handle the complexities of LiDAR point clouds as a means of addressing these challenges. In addition, advanced pre-processing techniques and data fusion methods are being investigated to reduce noise and improve the quality of the data.

#### 5.2.3. RQ3: Performance Evaluation Methods and Outcomes

The performance of clustering and MTT algorithms is evaluated using a variety of methods. The most common evaluation metrics are precision, recall, F1-score, and accuracy [102]. Typically, these metrics are used to evaluate various methods based on benchmark datasets, such as KITTI, nuScenes, and Waymo. The results of the reported methods vary depending on the algorithm and dataset used. The performance of learning-based methods, however, is generally better than that of other techniques, especially in complex environments where a large number of objects are present.

The purpose of this review study was to provide an overview of the current state of clustering and MTT methods for LiDAR point clouds used in autonomous driving systems. As part of our contributions, we categorize and identify various clustering and MTT methods, assess research gaps and challenges, and identify the challenges associated with these algorithms. As a result of this review, we hope to inform researchers and practitioners of the latest advancements in clustering and MTT methods for LiDAR point clouds, as well as encourage future research to address the remaining challenges in this field.

## 6. Conclusions

During this systematic review, we examined and assessed clustering and MTT methodologies for LiDAR point clouds. As a result of categorizing and reviewing existing research methodologies, we have gained an understanding of their potential as approaches to addressing the challenges inherent to the processing of LiDAR data. Among these difficulties are problems associated with data association and target identification, as well as dealing with noisy, sparse, and unorganized point cloud data. In light of the context of LiDAR point clouds, we identified opportunities for enhancement within the existing methodologies. Therefore, we have developed a deeper understanding of the current landscape of clustering and MTT algorithms for autonomous driving applications, as well as the challenges and limitations these approaches face.

In light of the insights revealed in this review, there is an urgent need to develop robust and efficient algorithms for processing LiDAR point clouds in the future. It will be increasingly important for autonomous vehicles to have a precise perception of their surroundings and to be able to locate themselves accurately as they evolve in dynamic and complex environments.

We intend to study the development of such models for LiDAR in order to enhance the perception capabilities of autonomous vehicles in the future. Furthermore, the generation of LiDAR point cloud datasets that accurately represent real-world complexity would enable us to evaluate and compare the performance of various clustering and MTT techniques.

The purpose of this review was not only to provide a comprehensive overview of the existing clustering and MTT algorithms for autonomous driving applications but also to outline potential research avenues. Our findings are expected to pave the way for further exploration of autonomous driving technology in this crucial area.

The findings of this study emphasize the need for more robust and efficient algorithms for the processing of LiDAR point clouds in the future. As autonomous vehicles continue to evolve, their success will increasingly depend on how accurately they are able to perceive their surroundings and locate themselves accurately in environments that are dynamic and complex.

As part of our future research, we will examine the development of such models for LiDAR to enhance the abilities of AVs perception. Moreover, the creation of LiDAR point clouds datasets that reflect the complexity of the real world would also allow us to evaluate and compare the performance of different clustering and MTT methods.

To conclude, this review has not only shed light on the existing landscape of clustering and MTT algorithms for autonomous driving applications but has also identified future research directions. Hopefully, our findings will serve as a stepping stone for future research in this critical area of autonomous driving technology.

## Figures and Tables

**Figure 1 sensors-23-06119-f001:**
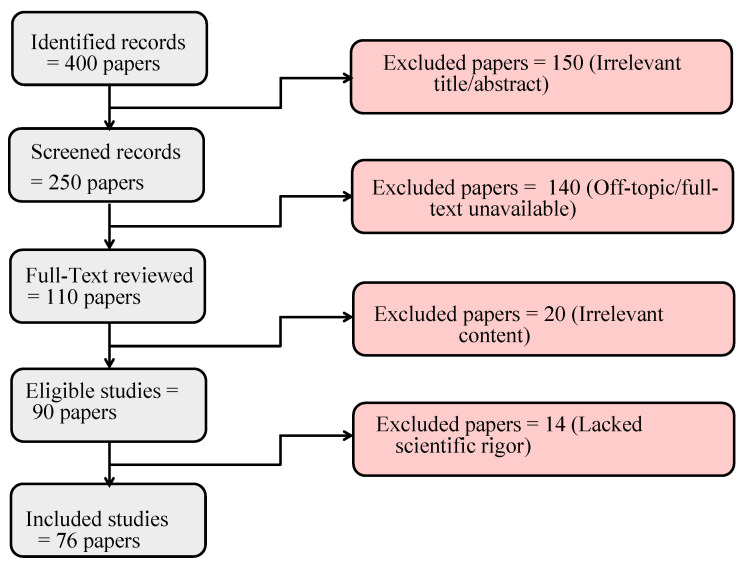
Systematic review flow diagram illustrating the stages involved. Studies are identified, screened, assessed for eligibility, and included, with reasons for exclusion detailed at each stage.

**Figure 2 sensors-23-06119-f002:**
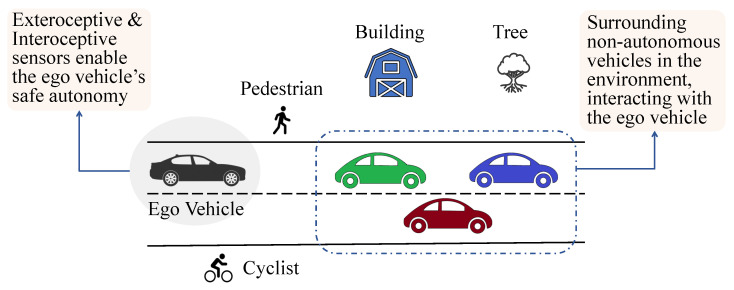
Ego vehicle equipped with advanced exteroceptive sensors for environmental perception and interoceptive sensors for monitoring its internal state, ensuring safe and efficient autonomous navigation.

**Figure 3 sensors-23-06119-f003:**
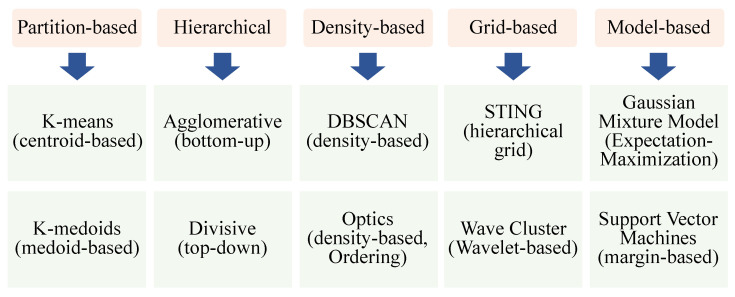
The different clustering techniques are presented in blocks accompanied by the relevant algorithms.

**Figure 4 sensors-23-06119-f004:**
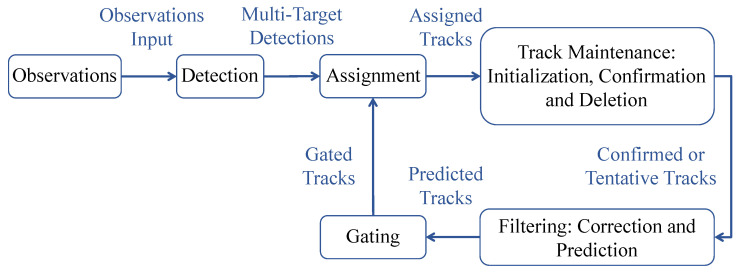
Components of Multi-target tracking.

**Table 1 sensors-23-06119-t001:** Our survey and recent research on point cloud data are summarized and compared. ‘#’ stands for ‘number’ and ‘refs.’ for ‘references’.

Survey	Year	Total # of Refs.	Clustering Taxonomy	# Clustering-Related Refs.	Multi-Target Tracking (MTT)	# MTT-Related Refs.	# Clustering-and-MTT-Related Refs.
Our work	2023	101	Covered	44	Covered	33	9
[29]	2022	150	Briefly covered	21	Not covered	1	0
[26]	2021	138	Not covered	30	Briefly covered	5	0
[23]	2021	47	Covered	24	Not covered	0	0
[22]	2020	256	Briefly covered	45	Briefly covered	9	1
[10]	2013	46	Covered	29	Not covered	0	0

**Table 2 sensors-23-06119-t002:** Frequency Distribution of Coverage for Multitarget Tracking (MTT), Clustering Taxonomy, and Clustering-MTT Related References in Reviewed Works. Cov. = Covered, Com. Cov. = Combine Covered, Br. Cov. = Briefly Covered, Not Cov. = Not Covered, Com. Tech. = Combine Techniques, MTT = Multitarget Tracking.

	MTT	Clustering	Clustering-MTT	Total
**Our work**				
Cov.	33	44	0	77
Com. Cov.	0	0	9	9
Total	33	44	9	86
[29]				
Br. Cov.	1	21	0	21
Not Cov.	0	0	0	0
Total	1	21	0	22
[26]				
Br. Cov.	5	30	0	35
Not Cov.	0	0	0	0
Total	5	30	0	35
[23]				
Not Cov.	0	0	0	0
Cov.	0	24	0	24
Total	0	24	0	24
[22]				
Br. Cov.	9	45	0	54
Com. Tech.	0	0	1	1
Total	9	45	1	55
[10]				
Not Cov.	0	0	0	0
Cov.	0	29	0	29
Total	0	29	0	29

**Table 3 sensors-23-06119-t003:** Pearson Chi-square test results for observed frequencies in each category.

	Pearson Chi-Square	Value	df	Significance (2-Sided)
Our Work		86.000	2	0.000
[29]		22.000	1	0.000
[26]		35.000	1	0.000
[23]		24.000	2	0.000
[22]		55.000	2	0.000
[10]		29.000	2	0.000

**Table 4 sensors-23-06119-t004:** List of important words and abbreviations.

Term/Abbreviation	Definition/Explanation
MTT	Multi-target tracking
LiDAR	Light Detection and Ranging
AVs	Autonomous Vehicles
FoV	Field of View
LSTM	Long-Short-Term Memory
MOT	Multi-Object-Tracking
DBSCAN	Density-Based Spatial Clustering of Applications with Noise
DNN	Deep Neural Network
DAC	DBSCAN-based adaptive clustering method
ROI	Region of Interest
WBLC	Window Based LiDAR Clustering
HEVC	High Efficiency Video Coding
CA	Constant Acceleration
MHT	Multiple Hypothesis Tracking
ICP	Iterative Closest Points
ADAS	Advanced Driver Assisted Systems
ACKF	Adaptive Cubature Kalman Filter
SRTs	Scale-Rotation-Translation score
DPCR	Dynamic Point Cloud Registration
EKF	Extended Kalman Filter
UKF	Unscented Kalman Filter
PF	Particle Filter
VLP-16	Velodyne LiDAR Pucks
MAR	Minimum Area Rectangle
JPDA	Joint Probabilistic Data Association
IMM	Interactive Multiple Model
PRISMA	Preferred Reporting Items for Systematic Reviews and Meta-Analyses
TLG	Two-Layer-Graph

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
