# Peer review of "Systematic and Comprehensive Review of Clustering and Multi-Target Tracking Techniques for LiDAR Point Clouds in Autonomous Driving Applications"

_sensors, 2023, doi:10.3390/s23136119_

Round 1

Reviewer 1 Report

In this study, the authors reviewed Clustering and Multi-Target Tracking Techniques for LiDAR Point Clouds in autonomous cars. The topic seems interesting. However, to improve the quality of the manuscript my suggestions are given below.

1) The problem statement should be mentioned earlier. The motivation for the study, key objectives, and contribution should be mentioned in bullets.

2)  In a review paper the comparison table is important. Please add a comparison in the introduction section and compare your own study and the most recent studies along with the pros and cons. Please have a look at this reference for making a comparison table. This paper is related to your topic.

Abbasi R, Mateen A, Ali Abid M, Khan S. A Step toward Next-Generation Advancements in the Internet of Things Technologies. Sensors. 2022; 22(20):8072.

3)The graph or image quality in this paper is very impressive.  Please improve the quality of all graphs in the final version.

4) future work should be mentioned in the collusion section. 

5) There is a need to add more recent references in the references section. It should be more than 74.

6) The discussion and future work should be merged.

The grammar typos and the English should be improved in the final version. 

Reviewer 2 Report

[Comment 1] Novelty

(lines 74-77) It is still unclear how the review studies differ from each other. The authors must compare the reviews with their study in a table for clarity.

[Comment 2] Review methodology

The authors must present a clear method on how the conducted their review to allow next researchers to reproduce the exact same review result with the same method, e.g., using http://prisma-statement.org/documents/PRISMA_2020_flow_diagram_new_SRs_v1.docx

[Comment 3] Writing quality and clarity

[Subcomment 3a] The authors must present the complete form of each abbreviation on its first use, e.g., MTT (once in the abstract, and once in the main text). It is impossible to understand what it means before reading Table 1.

[Subcomment 3b] Please ensure all texts in the figure are as large as the main text to allow readability.

[Subcomment 3c] "This study" at the first part of any paragraph is confusing, e.g., line 218. Please revise it.

Please refer to my review comments.

Reviewer 3 Report

Dear Editor,

Please find the attached document for comments.

Round 2

Reviewer 1 Report

The manuscript is in acceptable form.

It is fine.

Reviewer 2 Report

[Comment 1] Novelty

[Subcomment 1a] When comparing the studies in Figure 1, the authors must provide a quantitative comparison than a qualitative one.

[Subcomment 1b] The “briefly covered” classification for the “autonomous driving context” is very difficult to prove. Its definition is unclear.

[Comment 2] Review methodology

[Subcomment 2a] The PRISMA should contain the number of studies obtained in each step.

[Subcomment 2b] Also, the authors must add the PRISMA checklist, as requested in: https://www.mdpi.com/editorial_process#standards

Reviewer 3 Report

Please find the attached documents for comments.
